# The Spatial Distribution and Morphological Characteristics of Chum Salmon (*Oncorhynchus keta*) in South Korea

**Donghyun Hong [1], Gea-Jae Joo [1], Eunsong Jung [1], Jeong-Soo Gim [1], Ki Baik Seong [2], Doo-Ho Kim [3], Maurice J. M. Lineman [4], Hyun-Woo Kim [5] and Hyunbin Jo [1,6,*]**

[1] Department of Integrated Biological Science, Pusan National University, Busan 46241, Korea; hdh1201@pusan.ac.kr (D.H.); gjjoo@pusan.ac.kr (G.-J.J.); esongj0@pusan.ac.kr (E.J.); kjs1@pusan.ac.kr (J.-S.G.)
[2] Wonjin, Chungnam 32138, Korea; drsalmon@wonjin.biz
[3] Korea Fisheries Resources Agency, Dongmyeong-ro 119, Sonyang-myeon 25041, Korea; doho3206@fira.or.kr
[4] Department of Science, The Vantage Academy, NO.79 Yingze West Street, Taiyuan 030021, China; winterrider1967@gmail.com
[5] Department of Environmental Education, Sunchon National University, Suncheon 57922, Korea; hwkim@scnu.ac.kr
[6] Institute for Environment and Energy, Pusan National University, Busan 46241, Korea
\* Correspondence: prozeva@pusan.ac.kr; Tel.: +82-10-8807-7290

**Abstract:** Chum salmon (*Oncorhyncus keta*) is a cold-water species reported to migrate within a wide range of habitats, including Korea, Japan, North America, and Russia, playing important roles in the river–sea nutrient cycle and food web. However, research on this species has not been widely performed in South Korea owing to its geographical location at the southern edge of migration. In this study, we analyzed the spatial distribution and morphological characteristics of chum salmon migrating to South Korea using the length–weight relationship. We also analyzed 3 years of catch, sex ratio, and individual information (total length (cm), weight (kg), n = 4400) from ten rivers (eight in the East coast and two on the South coast) with a total of 17 years of water quality and the distance they traveled (n = 50) using multivariate analysis. As a result, we discovered a trend of less migration in the southern part of South Korea for all individuals migrating to South Korea. Furthermore, the weight ratio of males/females was significantly different ($p < 0.05$). Based on the length–weight relationship analysis, the *a* and *b* values were between 0.0011 and 0.038 and 2.65 and 3.49, respectively. In the correlation analysis, catch is negatively correlated with distance traveled and temperature ($p < 0.05$), and positively correlated with pH, dissolved oxygen, distance, and female ratio ($p < 0.05$). This is possibly the result of differences in water quality during early life stages or the presence of an estuarine barrage at the mouth of the Nakdong River. This research may be used as fundamental distribution and morphological variations of chum salmon in South Korea.

**Keywords:** Chum salmon; distribution patterns; length–weight relationship; water quality; sex ratio

## 1. Introduction

Salmonids comprise 11 genera, with 65–70 species worldwide [1]. The subfamily Salmonidae is composed of five to nine extant genera and approximately 30 species [2]. Among these, chum salmon (*Oncorhynchus keta*) is a cold-water fish distributed around the North Pacific Ocean including the Bering Sea, Alaska, and East Asia [3]. It is also understood that 95% of chum salmon biomass consists of oceanic organic matter, which enhances river–ocean nutrient circulation during decomposition following spawning [4]. Moreover, the behavior of returning from oceans to natal rivers to spawn is publicly well-known, rendering it aesthetically valuable [5]. For these reasons, the NPAFC (North Pacific Anadromous Fish Commission) has recognized the benefits of salmon for the ecosystem, and a significant amount of research on the lifecycles and resource management of Salmonidae has been conducted [6–9].

Chum salmon stock enhancement programs have been established by the North Pacific nations (Canada, Japan, Russia, South Korea, and so on) for representative migration-associated rivers and streams. These programs involve catching adult individuals for artificial fertilization and fry release [10,11], in order to obtain sustainable growth and breeding stock. By obtaining a relatively large number of chum salmon stock, studies on chum salmon in terms of the effect of spatial differences on morphological [12], genetic [13], and distribution differences [14] are globally documented. Species morphology and sex ratio, for instance, are an effective measure for identifying patterns and predictions for communities [15,16]; however, few studies of the spatial differences and morphology of chum salmon in South Korea have been conducted, owing to the limited migration of this species to South Korea, which is at the southern boundary of the distribution range in East Asia [17,18]. Additionally, research on the distribution and migration routes of chum salmon in South Korea has not been systematically performed; we hypothesize that those in Korea follow patterns similar to those of Japan and/or Russia [19]. Moreover, the return rate to South Korea has dropped significantly to nearly 1% owing to habitat fragmentation from dams and weir construction [20], high mortality at the juvenile stage [21], and a rise in ocean and river temperatures [22]. In addition, water quality deterioration is another important factor that reduces larval development and growth rates [23].

Therefore, the aims of the current research were to (1) identify distribution patterns of chum salmon migrating to South Korea; (2) localize regional morphological differences (body shape); (3) compare the sex ratio between chum salmon populations; and (4) quantify the relationship between water quality and the catch. In other words, we investigated the morphological differences between specimens from different regions in South Korea. This allows the determination of chum salmon distribution and morphological features affected by spatial differences in South Korea.

## 2. Materials and Methods

### 2.1. Study Area

This study was conducted in the south, east, and southeastern Korean peninsula region (Figure 1). In the ten rivers in the study, the eight uppermost rivers flow west to east, the second-lowest study station (St. 9, The Milyang River) flows to the southeast sea, and the lowermost study station (St. 10, the Seomjin River) flows to the southern sea of South Korea. These rivers are in temperate climate regions [24] with similar annual precipitation patterns, with over 60% of precipitation from June to September [25] due to the East Asian monsoon, and are driest in winter (November–January); consequently, hydrologic drought occurs from winter to early spring. Additionally, South Korea is an example of a tilted landform, where a chain of mountains is distributed along the eastern periphery of South Korea [26], making west-to-east rivers relatively short, small, and steep riverine systems. In addition, steep riverine systems on the eastern slope of South Korea often exhibit stream depletion, except in the wet season, while southward flowing rivers show a relatively low rate of stream depletion.

Currently, multi-purpose dams of different sizes exist in most South Korean rivers to prevent floods and droughts and to encourage eco-friendly public recreation [27]. In the study area, no artificial obstacles existed except for one on the Milyang River (St. 9). Furthermore, there is an estuarine barrage in the mouth of the Nakdong River, which was built in 1987 to satisfy human needs, such as salt intrusion prevention and a sustainable freshwater supply [28], and causing impoundment between the river and the sea [29]. The construction of the estuarine barrage has dramatic effects on fish communities [30], especially for migratory fishes [31]. Salmonid migration and behavior are closely related to environmental factors such as water velocity, topography, and water temperature [8,32,33], which possess the possibility of being affected by the estuarine barrage.

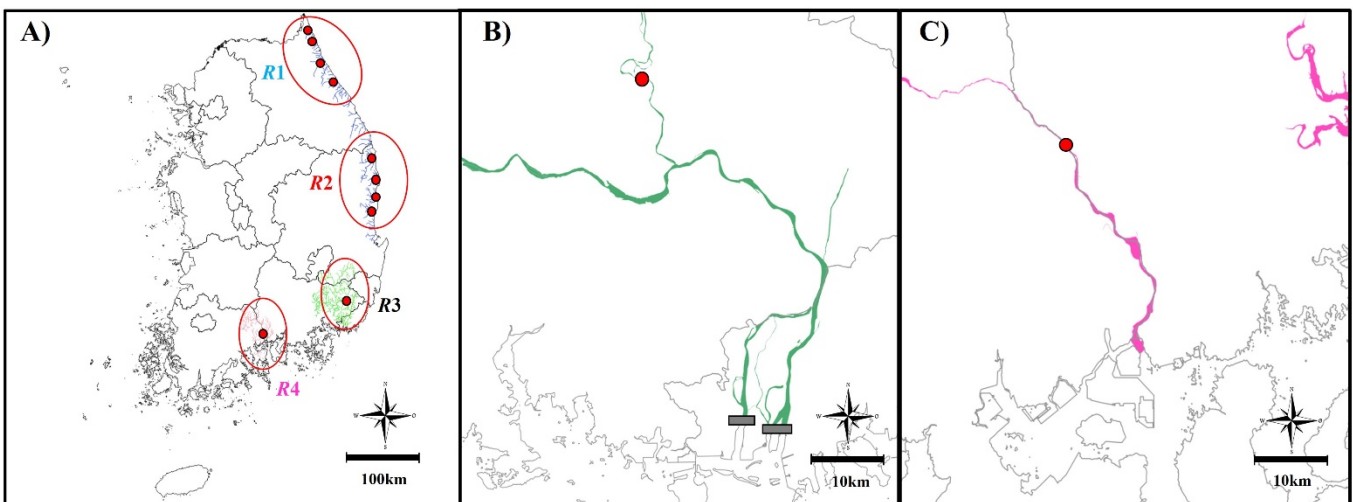

**Figure 1.** Distribution of study sites across South Korea. (**A**) In *R*1 area, St. 1, St. 2, St. 3, and St. 4 exist from the uppermost part towards the lower regions. *R*2 comprises St. 5 to St. 8 (St. 5 at the top), *R*3 comprises St. 9, and *R*4 comprises St. 10 (St. 1—the Myeong-pa stream, St. 2—the Buk stream, St. 3—the Namdae stream, St. 4—the Yeongok stream St. 5—the Song stream, St. 6—the Wang-pi stream, St. 7—the Pyeonghae-Namdae stream St. 8—the Osip stream, St. 9—the Milyang River, and St. 10—the Seomjin River). (**B**) Expansion of the *R*3 region; Grey-rectangles are estuarine barrages existing in the mouth of the Nakdong River. (**C**) Expansion of the *R*4 region.

## 2.2. Data Collection

All chum salmon data were collected from national salmon hatchery institutions. Data for chum salmon migrating to the St. 1–St. 4 rivers were collected by the Aquatic Living Resources Center of the East Sea (Uljin, South Korea), affiliated with the Korea Fisheries Resources Agency (FIRA); data for those migrating to the St. 5–St. 8 rivers were collected by the Gyeongsangbuk-do Research Center for Freshwater Fishes, also located in Uljin. Data collected in St. 9 were from the Gyeongsangnam-do Research Center for Freshwater Fish, located in Milyang. In addition, the data collected in St. 10 were from the Seomjingang River Fish Museum (Jeonranam-do), affiliated with the Institute of Ocean and Fisheries Science. The methodology of catching adult chum salmon is to install a series of fixed shore nets (5 × 5 cm mesh size; length, 80 m; width, 1.5 m) on rivers to which the chum salmon migrate to obstruct their migration. The nets guide them to artificial fishways heading to the center, where their sex was identified and eggs were collected. Each center accumulates the annual catch from each river and records information for each adult individual, such as total length (cm, ±0.1 cm), weight (kg, ±0.01 kg), and male/female rate. Individual sex was identified by comparing the mouth parts; males have a longer and sharper upper jaw and teeth, and females have a shorter, smoother upper jaw and teeth.

To obtain water quality data, we used the water environment information system [34]. We collected monthly data of pH, dissolved oxygen (DO, mg/L), biological oxygen demand (BOD, mg/L), total nitrogen (TN, mg/L), total phosphorus (TP, mg/L), water temperature (°C), and conductivity (μS/cm). We utilized water quality data from 2014 to 2019 for *R*1 (n = 9), and 2004 to 2020 for *R*2 (n = 24) and *R*4 (n = 17). Water quality in *R*3 was excluded owing to insignificant and insufficient data.

## 2.3. Data Analysis

We hypothesized that chum salmon in South Korea will display significantly different mean values for total length (cm) and weight (kg) based on sex. To analyze the spatial morphology differences in accordance with regions and sexes, we used an independent *t*-test on mean total length (cm) and weight (kg) on different sexes using SPSS25. The dataset used in this analysis consists of St. 1, St. 2, and St. 3 in 2018; St. 5 in 2018–2020; and St. 9 and St. 10 in the year 2020.

We also performed calculations of length–weight relationships (LWRs) introduced by Froese [35] on chum salmon in South Korea. The formula for the LWR is as follows:

$$W = aL^b, \tag{1}$$

(W: weight in grams; L: length in cm; *a* and *b*: parameters).

We used the natural logarithm (ln) on both sides and constructed a linear regression model using SigmaPlot 10.0. The transformed LWR formula is shown below:

$$\ln W = \ln a + b \ln L, \tag{2}$$

(W: weight in grams; L: length in cm; *a* and *b*: parameters).

We then eliminated outliers until the coefficient of determination ($r^2$) reached 0.95. Additionally, to analyze differences in sex ratio in accordance with the regions, a one-way analysis of variance (one-way ANOVA) was carried out using SPSS 25. We subdivided ten study stations into four regional groups: *R1*, *R2*, *R3*, and *R4* (*R1* = St. 1–4, *R2* = St. 5–8, *R3* = St. 9, and *R4* = St. 10), and then utilized a one-way ANOVA for the data from the four regions. Statistical significance was set at $p < 0.05$. We excluded years for which accurate sex ratio data were not available and utilized sex ratio data only from five years (2014–2016, 2018, and 2019) for *R1* and 17 years (2004–2020) for *R2* and *R4*. *R3* region is excluded owing to insignificant data (only the year 2020 is available).

To determine the correlation between chum salmon distribution and sex ratio and water quality, we performed a principal component analysis (PCA) using R version 4.0.5. We specifically collected monthly water quality data from September to November to match migration times to freshwater, and then averaged them to yield the annual data. Apart from the correlation between distribution, sex ratio, and water quality, we also used regional distances measured from the uppermost stream (St. 1) as another environmental parameter.

## 3. Results

### 3.1. Morphological Characteristic Differences

Among over 120,000 chum salmons that migrated to South Korea within seventeen years, we were able to calculate the mean total length and body weights of chum salmon in South Korea using individual 4400 chum salmons whose body information was recorded. The mean total length was 68.49 cm (±5.4, standard deviation) and 2.66 kg (±0.71). A total of 2070 males and 2330 females from six rivers (St. 1, St. 2, St. 3, St. 6, St. 9, and St. 10) were used to identify differences in total length and body mass. Females that migrated to South Korea had a mean total length of 68.42 cm (±4.7), while males were 68.55 cm (±5.9) on average, showing no significant differences ($p > 0.05$, Table 1). On the other hand, the weight for females was calculated to be 2.69 kg (±0.67) and 2.65 kg (±0.75) for males, which was significantly different related to the sex ($p < 0.05$, Table 1).

**Table 1.** Mean length and mean weight of chum salmon (*Oncorhynchus keta*) surveyed in South Korea (* *t*-test, $p < 0.05$).

| Sex | Total Number of Data (*n*) | Mean Length (cm) ±Standard Deviation | Mean Weight (Kg) ±Standard Deviation |
|---|---|---|---|
| Total | 4400 | 68.49 ± 5.4 | 2.66 ± 0.71 |
| Male | 2070 | 68.55 ± 5.9 | * 2.65 ± 0.75 |
| Female | 2330 | 68.42 ± 4.7 | * 2.69 ± 0.67 |

The LWRs of chum salmon during all study periods were calculated. For individuals migrating to South Korea, the *a* value was 0.0047 (±0.0009) and the *b* value was 3.129 (±0.023) (n = 2074). The *a* and *b* values differed based on the sex: 0.0038 (±0.00056) and 3.18 (±0.038), respectively, for females; 0.0052 (±0.0007) and 3.1 (±0.035), respectively, for males. Annual *a* and *b* values were calculated for the year 2018 from St. 1 (n = 18), St. 2 (n = 11), and St. 3 (n = 432). We also calculated the LWRs from St. 6 (n = 1384) in 2018,

2019, and 2020, as well as St. 9 (n = 49) and St. 10 (n = 386) in 2020. Specifically, specimens sampled at St. 1, St. 2, and St. 3 showed *a* values of 0.0011 ($\pm$0.004), 0.004 ($\pm$0.001), and 0.0013 ($\pm$0.0002), respectively. St. 6 showed an average *a* value of 0.009 $\pm$ 0.001. In addition, St. 9 and St. 10 showed *a* values of 0.038 ($\pm$0.017) and 0.005 ($\pm$0.001), respectively (Figure 2). Chum salmon from St. 1, St. 2, and St. 3 showed *b* values of 3.45 ($\pm$0.05), 3.17 ($\pm$0.08), and 3.49 ($\pm$0.52), respectively. In addition, chum salmon from St. 6, St. 9, and St. 10 showed average *b* values of 2.97 ($\pm$0.03), 2.65 ($\pm$0.14), and 3.1 ($\pm$0.07), respectively (Figure 3).

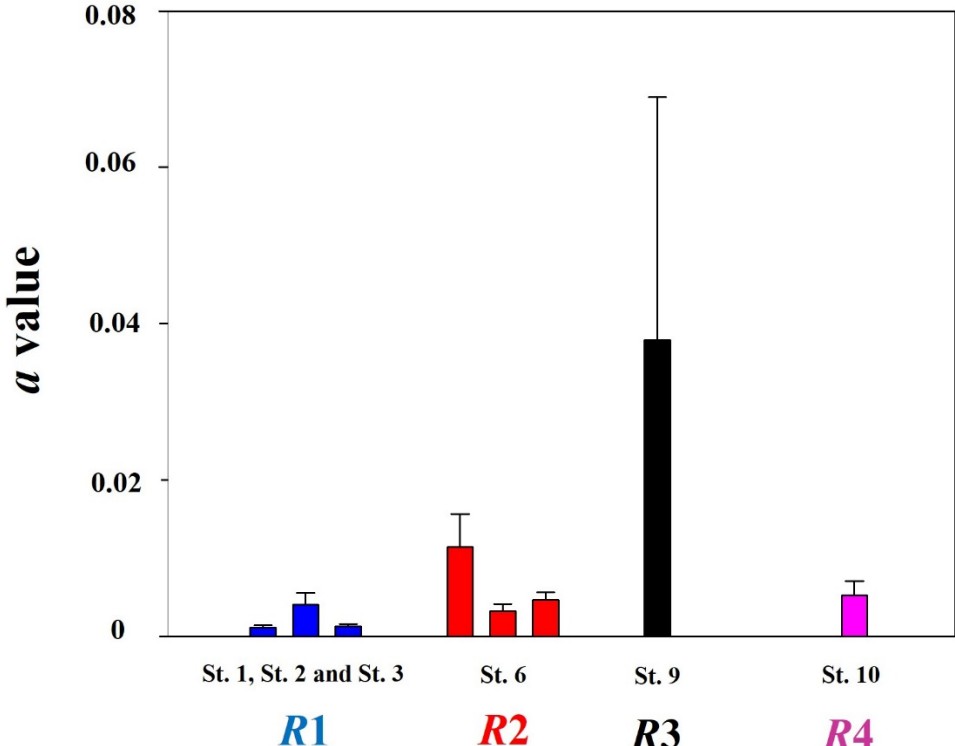

**Figure 2.** Regional *a* values for chum salmon (*Oncorhynchus keta*) individuals that migrated to South Korea from 2018 to 2020. The *a* value showed the greatest value on the *R*3 region, while the *R*1 region showed the smallest value. For the *a* values, we calculated them using datasets in the year 2018 for *R*1 regions. Datasets from 2018 to 2020 were used to calculate the *a* values in *R*2, while datasets in 2020 were used for *R*3 and *R*4 regions (n = 422 for St. 1; 11 for St. 2; 18 for St. 3; 258 for St. 6 in 2018, 566 in 2019, and 883 in 2020. For St. 9 and St. 10, the 'n' is 43 and 291, respectively).

### 3.2. Regional Sex Ratio Differences

The sex ratio data from St. 1, St. 3, and St. 4 obtained over a period of five years were used. In addition, data obtained from St. 2 for four years were used. We grouped these four streams into the *R*1 region, as they were relatively close to each other. Data regarding the sex of chum salmon from St. 6 and St. 10 obtained over a 17-year period were used to calculate the sex ratio and were termed R2 and R4, respectively. For the *R*1 region, the proportion of females was 44.66% ($\pm$7.619), while the proportions of females in the other two regions (*R*2 and *R*4) were 41.95% ($\pm$4.53) and 39.79% ($\pm$7.585), respectively (Tables A1–A4).

We used a one-way ANOVA to examine the significance of the sex ratio in the three regions. St. 9 was not included because of insufficient data (n = 1). None of the regions showed significant differences in sex ratio ($p > 0.05$).

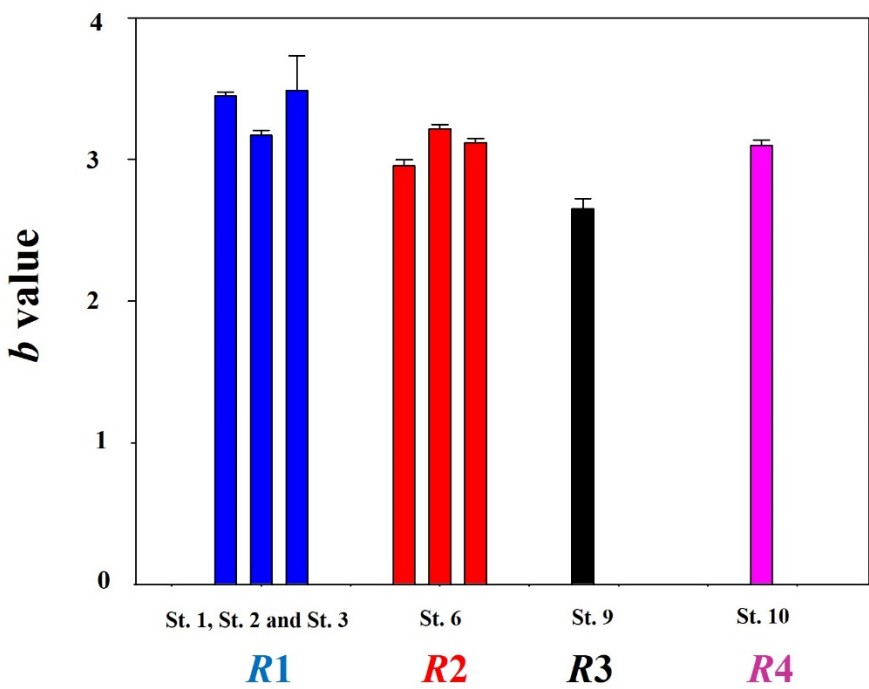

**Figure 3.** Regional *b* values for chum salmon (*Oncorhynchus keta*) individuals that migrated to South Korea from 2018 to 2020. Unlike Figure 2, the *R*1 region showed the greatest *b* value among four regions, while *R*3 showed the smallest value (n = 422 for St. 1; 11 for St. 2; 18 for St. 3; 258 for St. 6 in 2018, 566 in 2019, and 883 in 2020. For St. 9 and St. 10, the 'n' is 43 and 291, respectively).

*3.3. Correlation of Distribution with Water Quality*

We first analyzed the regional water quality values and how they differ between the areas. The seven water quality parameters (temperature, conductivity, TP, TN, BOD, pH, and DO) are presented in Table A5. All parameters, except for TN, were significantly different in accordance with regional differences (Table A6). Specifically, for *R*1, pH and DO are significantly higher than other regions, while other values are mostly lower than other regions. To evaluate the correlations of chum salmon migration numbers with each parameter, all seven parameters were selected for PCA (Figure 4). In addition, the regional catch, female ratio, and regional distance were selected as inherent values of each individual. In the ordination, 39.9% of the variation was related to axis 1 (PC1), while axis 2 (PC2) explained 17.4% (total 57.3% of variance) in the biplot. Each vector indicates the direction and strength of each environmental variable relative to the overall interrelations. In Figure 4, distance, TP, and temperature have a relatively strong positive correlation to PC1, while BOD and TN show a relatively strong negative correlation to PC2. For vectors presented in the PCA, the catch was positively correlated with pH ($p < 0.01$), female ratio ($p < 0.05$), and DO ($p < 0.05$), while it was negatively correlated with temperature ($p < 0.05$) and distance ($p < 0.01$, Table 2). Each colored number (1–44) in the biplot represents different physiochemical properties, including water quality and female ratio. Physiochemical properties in *R*1 (1–9) seem to be assembled in the bottom-left side of the biplot, where relatively high positive correlations with catch, pH, DO, and female ratio and negative correlations with temperature, distance, conductivity, and TP can be observed. Physiochemical properties from *R*2 (10–26) are more or less gathered in the top-middle side of the biplot, showing relatively strong negative correlations with BOD and TN. However, the explanatory power of the *R*2 region using PC1 is relatively low. Though there is only one data from *R*3, physiochemical properties from *R*3 and *R*4 are relatively located together, showing positive correlations with temperature, distance, conductivity, and TP.

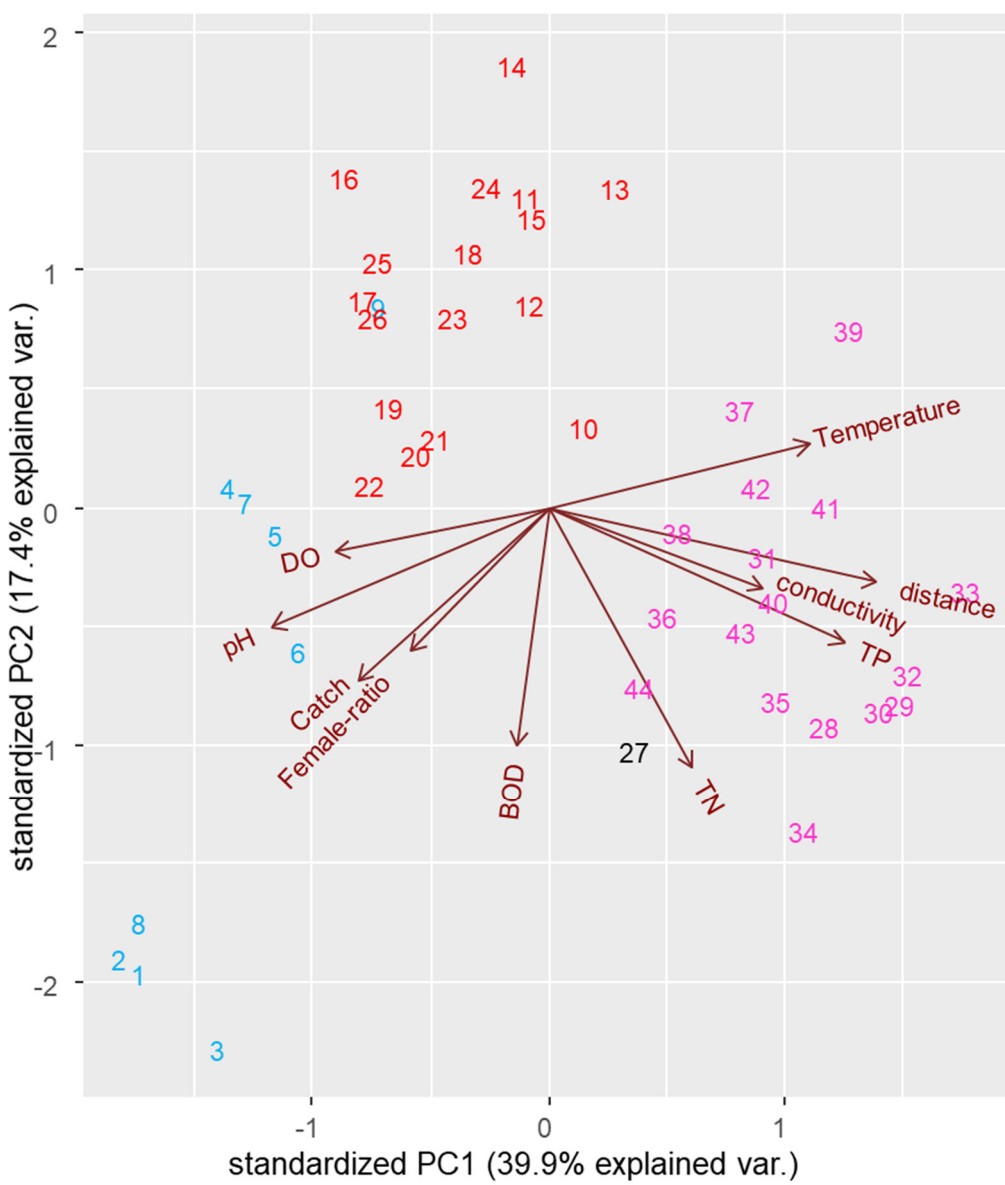

**Figure 4.** Principal component analysis (PCA) of ten variables (catch, female ratio, dissolved oxygen (DO), pH, biological oxygen demand (BOD), total nitrogen (TN), total phosphorus (TP), conductivity, distance, and temperature). The parameterized stations (1–9: *R1*, 10–26: *R2*, 27: *R3*, 28–44: *R4*) were gathered in accordance with three regions (*R1*, *R2*, *R4*).

**Table 2.** The Pearson correlation coefficient and significance (*p*) between the variables. Significance at * $p < 0.05$ and ** $p < 0.01$.

| | Catch | Female Ratio | pH | DO | BOD | TN | TP | Temp. | Conductivity | Distance |
|---|---|---|---|---|---|---|---|---|---|---|
| **Catch** | 1.000 | 0.352 * | 0.429 ** | 0.369 * | 0.223 | 0.075 | −0.284 | −0.300 * | −0.164 | −0.397 ** |
| | | 0.019 | 0.004 | 0.014 | 0.145 | 0.626 | 0.062 | 0.048 | 0.288 | 0.008 |
| **Female Ratio** | 0.352 * | 1.000 | 0.427 ** | 0.096 | 0.011 | 0.112 | −0.185 | −0.202 | −0.100 | −0.293 |
| | 0.019 | | 0.004 | 0.535 | 0.943 | 0.469 | 0.230 | 0.189 | 0.520 | 0.054 |
| **pH** | 0.429 ** | 0.427 ** | 1.000 | 0.344 * | 0.307 * | −0.109 | −0.479 ** | −0.573 ** | −0.349 * | −0.672 ** |
| | 0.004 | 0.004 | | 0.022 | 0.043 | 0.479 | 0.001 | 0.000 | 0.020 | 0.000 |
| **DO** | 0.369 * | 0.096 | 0.344 * | 1.000 | 0.056 | −0.136 | −0.333 * | −0.562 ** | −0.302 * | −0.397 ** |
| | 0.014 | 0.535 | 0.022 | | 0.717 | 0.378 | 0.027 | 0.000 | 0.047 | 0.008 |
| **BOD** | 0.223 | 0.011 | 0.307 * | 0.056 | 1.000 | 0.256 | 0.095 | −0.146 | 0.116 | −0.020 |
| | 0.145 | 0.943 | 0.043 | 0.717 | | 0.093 | 0.541 | 0.343 | 0.453 | 0.898 |

**Table 2.** *Cont.*

|  | Catch | Female Ratio | pH | DO | BOD | TN | TP | Temp. | Conductivity | Distance |
|---|---|---|---|---|---|---|---|---|---|---|
| **TN** | 0.075 | 0.112 | -0.109 | −0.136 | 0.256 | 1.000 | 0.555 ** | 0.113 | 0.133 | 0.566 ** |
|  | 0.626 | 0.469 | 0.479 | 0.378 | 0.093 |  | 0.000 | 0.465 | 0.389 | 0.000 |
| **TP** | −0.284 | −0.185 | −0.479 ** | −0.333 * | 0.095 | 0.555 ** | 1.000 | 0.475 ** | 0.562 ** | 0.852 ** |
|  | 0.062 | 0.230 | 0.001 | 0.027 | 0.541 | 0.000 |  | 0.001 | 0.000 | 0.000 |
| **Temp.** | −0.300 * | −0.202 | −0.573 ** | −0.562 ** | −0.146 | 0.113 | 0.475 ** | 1.000 | 0.319 * | 0.574 ** |
|  | 0.048 | 0.189 | 0.000 | 0.000 | 0.343 | 0.465 | 0.001 |  | 0.035 | 0.000 |
| **Conductivity** | −0.164 | −0.100 | −0.349 * | −0.302 * | 0.116 | 0.133 | 0.562 ** | 0.319 * | 1.000 | 0.520 ** |
|  | 0.288 | 0.520 | 0.020 | 0.047 | 0.453 | 0.389 | 0.000 | 0.035 |  | 0.000 |
| **Distance** | −0.397 ** | −0.293 | −0.672 ** | −0.397 ** | −0.020 | 0.566 ** | 0.852 ** | 0.574 ** | 0.520 ** | 1.000 |
|  | 0.008 | 0.054 | 0.000 | 0.008 | 0.898 | 0.000 | 0.000 | 0.000 | 0.000 |  |

## 4. Discussion

The distribution of chum salmon in South Korea is principally concentrated on *R*1 (Tables A1–A4), and this seems to be due to the large number of offspring released in the *R*1 region [10]. Ruggerone and Irvine [36] calculated that approximately 70% of chum salmon individuals returning to Asia are from artificial hatchery production; therefore, the number of offspring produced by artificial fertilization could be the most crucial factor in the total catch. Furthermore, relatively high genetic diversity due to a large number of males/females could prevent the Allee effect in the *R*1 region.

Regional differences in the sex ratio of chum salmon individuals were not significant. It is likely that the insignificance is due to the relatively small study area in each region (*R*1–*R*4). Although there was no significant regional difference in the female ratio, we were able to see a positive correlation with the catch. Brykov et al. [37] documented that the female ratio in the Kamchatka River showed a negative correlation with spawner density, which differs from our results. For sex determination of sockeye salmon (*O. nerka*), the temperature can alter their sex in the early life history stages [38]. Thus, the same strategy could be applied to chum salmon, but no related data are available. Moreover, we suppose that other salmon variables such as migration age and early life stage diet could be potential factors in sex determination. So, there needs to be additional data collection at national salmon hatchery institutions for further research. No significant difference was found in the total length, yet we found a significant difference in their weights ($p < 0.05$). A significant difference in weight may be attributed to the presence of eggs in the female.

In the PCA, catch is positively correlated with DO and pH, while it is negatively correlated with temperature and distance (Figure 4, Table 2). In the early stages of the life history of salmon, a high DO [39,40] and a low temperature [41,42] is particularly important for water quality and have been well investigated in terms of early mortality based on past research. In addition, low pH results in the disappearance of fish populations as a result of water quality reducing site use or choice [43,44], as well as for chum salmon [45]. Therefore, *R*2, *R*3, and *R*4 regions may not be able to hold large numbers of the species owing to relatively polluted water quality parameters, such as low DO and a relatively small number of fry released. We also observed high conductivity in *R*4, and it is thought to be due to the water quality measuring station located in a brackish-zone [46], despite its rather inland location. Changes in water quality (often caused by global warming), such as DO or temperature, can possibly cause a more severe decrease in abundance in Korea than in other countries, as Korea lay on the southern edge of chum salmon's migration. Specifically, for water temperature, we observed that the mean temperature in all regions was about 15–18 °C (Table A5). Chum salmons have an acute preference for temperatures of 7 °C to 11 °C [3], which is considerably lower than referred regions. As global warming proceeded [47], Japan has already experienced a decrease in the carrying capacity and loss of migration routes of chum salmons [48,49], which gives us the possibility that carrying capacity in South Korea is also, or more severely, decreased.

To the best of our knowledge, this study represents the assessment of spatial morphological characteristics of chum salmon in South Korea, including a study carried out by Myeong and Kim [50], in which LWR data were excluded. In addition, research on the morphology of chum salmon has focused on juvenile [51], fish scales [52], and otoliths [53,54] globally. Froese et al. [55] used a Bayesian approach for estimating LWRs in fish and subdivided the fish body shape into four types: eel-like, elongated, fusiform, and short and deep. Analysis revealed that fish with an elongated body shape exhibit an $a$ value of 0.0018–0.00842 at the 95% confidence interval and an $a$ value of 0.00514–0.0245 for fusiform fish. As individuals migrating to South Korea exhibited $a$ values in the range of 0.0011–0.038, it is expected that those in South Korea have a more or less 'blended' body shape with elongated and fusiform structures. Fish are the most susceptible vertebrates to environmentally induced morphological variations, demonstrating greater variance within and between populations [56,57]. Thus, environmental variables such as prey composition in early life stages or genetic variation may alter the $a$ value of chum salmon in South Korea. In addition, it is a global phenomenon that its weight is in a gradual decline [36], so it may affect the $a$ value. Likewise, research on the $b$ value of chum salmon has not been well documented. From the information presented in FishBase [58], it is shown that the $b$ value of chum salmon varies from 2.93 to 3.25, which establishes the $b$ values from St. 9 (2.65), St. 1 (3.45), and St. 3 (3.49) as outliers. The low $b$ values for St. 9 could be attributed to the presence of the Nakdong estuary barrage. Many studies have observed that, in the presence of an estuary barrage and water impoundment, salmon migration is often delayed and obstructed [59], due to increased residence time [60], loss of direction [61], and susceptibility to disease [62]. Moreover, data were collected rather upstream (Figure 1) for St. 9 and St. 10. They cease foraging during their migration and have to rely on stored energy [63], so upstream migration in a river may have decreased the $b$ value. Alternatively, the relatively high values for St. 1 and St. 3 (3.45 and 3.49, respectively) could be attributed to relatively short migration distances and higher water quality in early life history stages. In addition, the $b$ value for St. 3 may be inaccurate owing to the lack of available data (n = 18). So far, not enough available data for Korean chum salmon's length–weight relationship are presented, which implies demands for further research to compare the regional morphologies of South Korea.

Thus far, we have performed an analysis of data on the distribution and morphological characteristics of chum salmon in South Korea. They possess a large geographic distribution [64], while their body weight exhibits more or less fluctuating patterns [65], indicating that the long-term monitoring of migration to South Korea is vital. However, to fully discover migration patterns and morphological characteristics, further data regarding chum salmon migration to South Korea are required. In addition, along with the size, spawn age is also an important life-history trait [66]; thus, measuring the age of individuals of this species during migration would also be a focus for further research. Moreover, more detailed consideration of the geographical traits of South Korea and global warming is necessary, as released offspring often suffer high mortality due to increased sea temperatures [67], which would be fatal for offspring released to southern rivers ($R$3 and $R$4). In this point of view, though marine water quality parameters (SST, sea surface temperature and SSS, sea surface salinity, among others) play an important role in the catch [68], we only utilized inland water quality data, which indeed leaves a limitation to fully discover the correlation of water quality with the catch. The return rate of chum salmon in South Korea is thought to be around 1% [10], which is relatively low compared with that in Japan (~5%, [11]). To understand the mechanisms underlying the migration of its offspring, fish preserves for immature individuals, as well as the maintenance of good water quality, should be established first.

## 5. Conclusions

Chum salmon (*O. keta*) are found circumpolar in their distribution; however, the South Korean peninsula marks the southern edge of its migration range. Results show that the

sex ratios showed little variation, while morphological characteristics did show significant variation relative to the location within the sampling area. Also, there was a strong relationship between water quality and developmental growth (the *a* value). As well, DO and pH showed a positive relationship to catch while temperature and distance had a negative relationship. The more polluted the water the lower the diversity and alterations to the temperature will also impact the species morphology and sex ratios. Results also show that the long-term monitoring of the Chum Salmon population in South Korea is important to determine how further changes in water quality and climate will affect the population within South Korea. These changes will require further study to determine the effects on the Chum Salmon population.

**Author Contributions:** Conceptualization, D.H. and H.J.; methodology, D.H., E.J. and H.J.; data curation D.-H.K., K.B.S. and H.-W.K.; validation, M.J.M.L. and H.-W.K.; resources, D.-H.K.; writing—original draft preparation, D.H., E.J. and J.-S.G.; writing—review and editing, D.H., G.-J.J. and K.B.S.; supervision, J.-S.G., K.B.S., D.-H.K., G.-J.J., M.J.M.L., H.-W.K. and H.J. All authors have read and agreed to the published version of the manuscript.

**Funding:** This research was funded by the National Research Foundation of Korea, grant number NRF-2020R1C1C1009066.

**Data Availability Statement:** The datasets used during and/or analyzed during the current study are available from the corresponding authors on reasonable request.

**Conflicts of Interest:** The authors declare no conflict of interest.

## Appendix A

**Table A1.** The catch of chum salmon (*Oncorhynchus keta*) individuals and the number of females (female ratio) in the *R*1 region.

| Region | St. Number | Year | Catch | The Number of Female | Female Ratio |
|--------|-----------|------|-------|----------------------|--------------|
| *R1* | St. 1 | 2014 | 2214 | 919 | 0.4151 |
| *R1* | St. 1 | 2015 | 532 | 245 | 0.4605 |
| *R1* | St. 1 | 2016 | 1564 | 831 | 0.5313 |
| *R1* | St. 1 | 2018 | 1627 | 772 | 0.4745 |
| *R1* | St. 1 | 2019 | 166 | 56 | 0.3373 |
| *R1* | St. 2 | 2014 | 1638 | 718 | 0.4383 |
| *R1* | St. 2 | 2016 | 1370 | 634 | 0.4628 |
| *R1* | St. 2 | 2018 | 3430 | 1382 | 0.4029 |
| *R1* | St. 2 | 2019 | 1057 | 364 | 0.3444 |
| *R1* | St. 3 | 2014 | 31,833 | 15,261 | 0.4794 |
| *R1* | St. 3 | 2015 | 18,151 | 9108 | 0.5018 |
| *R1* | St. 3 | 2016 | 11,262 | 6342 | 0.5631 |
| *R1* | St. 3 | 2018 | 6635 | 3010 | 0.4537 |
| *R1* | St. 3 | 2019 | 2790 | 1208 | 0.433 |
| *R1* | St. 4 | 2014 | 4011 | 2139 | 0.5333 |
| *R1* | St. 4 | 2015 | 4249 | 2062 | 0.4853 |
| *R1* | St. 4 | 2016 | 3462 | 1674 | 0.4835 |
| *R1* | St. 4 | 2018 | 1971 | 876 | 0.4444 |
| *R1* | St. 4 | 2019 | 457 | 110 | 0.2407 |

**Table A2.** The catch of chum salmon (*Oncorhynchus keta*) individuals and the number of females (Female ratio) in the *R*2 region.

| Region | St. Number | Year | Catch | The Number of Female | Female Ratio |
|--------|-----------|------|-------|----------------------|--------------|
| *R2* | St. 6 | 2004 | 454 | 191 | 0.4229 |
| *R2* | St. 6 | 2005 | 205 | 77 | 0.3756 |

**Table A2.** *Cont.*

| Region | St. Number | Year | Catch | The Number of Female | Female Ratio |
|--------|-----------|------|-------|---------------------|--------------|
| *R2* | St. 6 | 2006 | 1221 | 534 | 0.4373 |
| *R2* | St. 6 | 2007 | 1615 | 691 | 0.4279 |
| *R2* | St. 6 | 2008 | 375 | 135 | 0.36 |
| *R2* | St. 6 | 2009 | 706 | 266 | 0.3768 |
| *R2* | St. 6 | 2010 | 1145 | 504 | 0.4402 |
| *R2* | St. 6 | 2011 | 727 | 365 | 0.5021 |
| *R2* | St. 6 | 2012 | 1286 | 541 | 0.4207 |
| *R2* | St. 6 | 2013 | 1286 | 561 | 0.4362 |
| *R2* | St. 6 | 2014 | 2091 | 881 | 0.4213 |
| *R2* | St. 6 | 2015 | 1339 | 522 | 0.3898 |
| *R2* | St. 6 | 2016 | 1077 | 488 | 0.4531 |
| *R2* | St. 6 | 2017 | 1136 | 451 | 0.397 |
| *R2* | St. 6 | 2018 | 442 | 145 | 0.3281 |
| *R2* | St. 6 | 2019 | 881 | 391 | 0.4438 |
| *R2* | St. 6 | 2020 | 1651 | 822 | 0.4979 |

**Table A3.** The catch of chum salmon (*Oncorhynchus keta*) individuals and the number of females (Female ratio) in the *R*3 region.

| Region | St. Number | Year | Catch | The Number of Sex-Identified Individuals | Female Ratio |
|--------|-----------|------|-------|------------------------------------------|--------------|
| *R3* | St. 9 | 2020 | 197 | 61/95 | 0.391 |

**Table A4.** The catch of chum salmon (*Oncorhynchus keta*) individuals and the number of females (Female ratio) in the *R*4 region.

| Region | St. Number | Year | Catch | The Number of Female | Female Ratio |
|--------|-----------|------|-------|---------------------|--------------|
| *R4* | St. 10 | 2004 | 163 | 75 | 0.4601 |
| *R4* | St. 10 | 2005 | 243 | 102 | 0.4198 |
| *R4* | St. 10 | 2006 | 387 | 165 | 0.4264 |
| *R4* | St. 10 | 2007 | 419 | 179 | 0.4272 |
| *R4* | St. 10 | 2008 | 68 | 29 | 0.4265 |
| *R4* | St. 10 | 2009 | 96 | 37 | 0.3854 |
| *R4* | St. 10 | 2010 | 54 | 27 | 0.5 |
| *R4* | St. 10 | 2011 | 59 | 30 | 0.5085 |
| *R4* | St. 10 | 2012 | 79 | 36 | 0.4557 |
| *R4* | St. 10 | 2013 | 162 | 57 | 0.3519 |
| *R4* | St. 10 | 2014 | 188 | 61 | 0.3245 |
| *R4* | St. 10 | 2015 | 208 | 48 | 0.2308 |
| *R4* | St. 10 | 2016 | 124 | 36 | 0.2903 |
| *R4* | St. 10 | 2017 | 265 | 88 | 0.3321 |
| *R4* | St. 10 | 2018 | 704 | 239 | 0.3395 |
| *R4* | St. 10 | 2019 | 692 | 316 | 0.4566 |
| *R4* | St. 10 | 2020 | 569 | 244 | 0.4288 |

**Table A5.** Water qualities measured in each region.

| Parameters | Regions | Average | Standard Deviation | Standard Error | n |
|------------|---------|---------|--------------------|----------------|---|
| | *R1* | 8.95 | 0.46 | 0.15 | 9 |
| **pH** | *R2* | 7.83 | 0.36 | 0.07 | 24 |
| | *R4* | 7.57 | 0.19 | 0.05 | 17 |



**Table A5.** *Cont.*

| Parameters | Regions | Average | Standard Deviation | Standard Error | n |
|---|---|---|---|---|---|
| **DO (mg/L)** | *R1* | 10.79 | 1.16 | 0.39 | 9 |
| | *R2* | 9.62 | 1.12 | 0.23 | 24 |
| | *R4* | 9.28 | 1.19 | 0.29 | 17 |
| **BOD (mg/L)** | *R1* | 0.93 | 0.29 | 0.1 | 9 |
| | *R2* | 0.68 | 0.22 | 0.04 | 24 |
| | *R4* | 0.75 | 0.19 | 0.05 | 17 |
| **TN (mg/L)** | *R1* | 1.25 | 0.36 | 0.12 | 9 |
| | *R2* | 1.3 | 0.96 | 0.196 | 24 |
| | *R4* | 1.56 | 0.29 | 0.071 | 17 |
| **TP (mg/L)** | *R1* | 0.014 | 0.003 | 0.001 | 9 |
| | *R2* | 0.017 | 0.009 | 0.002 | 24 |
| | *R4* | 0.043 | 0.012 | 0.003 | 17 |
| **Temperature (°C)** | *R1* | 15.64 | 1.78 | 0.595 | 9 |
| | *R2* | 17.77 | 1.87 | 0.38 | 24 |
| | *R4* | 18.76 | 1.14 | 0.28 | 17 |
| **Conductivity (μS/cm)** | *R1* | 227.63 | 119.67 | 39.89 | 9 |
| | *R2* | 154.61 | 54.05 | 11.03 | 24 |
| | *R4* | 4534.29 | 5326.83 | 1291.95 | 17 |

**Table A6.** One-way ANOVA results in accordance with different regions.

| Parameters | Source of Variance | Sum of Squares | df | Mean Square | F | Significance |
|---|---|---|---|---|---|---|
| **pH** | **Between groups** | 11.72 | 2 | 5.86 | 53.31 | <0.01 |
| | **Within group** | 5.17 | 47 | 0.11 | | |
| | **Total** | 16.89 | 49 | | | |
| **DO (mg/L)** | **Between groups** | 13.72 | 2 | 6.86 | 5.18 | <0.01 |
| | **Within group** | 62.22 | 47 | 1.32 | | |
| | **Total** | 75.95 | 49 | | | |
| **BOD (mg/L)** | **Between groups** | 0.40 | 2 | 0.20 | 3.98 | <0.05 |
| | **Within group** | 2.37 | 47 | 0.05 | | |
| | **Total** | 2.77 | 49 | | | |
| **TN (mg/L)** | **Between groups** | 0.85 | 2 | 0.42 | 0.85 | 0.44 |
| | **Within group** | 23.55 | 47 | 0.50 | | |
| | **Total** | 24.39 | 49 | | | |
| **TP (mg/L)** | **Between groups** | 0.008 | 2 | 0.004 | 47.13 | <0.01 |
| | **Within group** | 0.004 | 47 | <0.001 | | |
| | **Total** | 0.012 | 49 | | | |
| **Temperature (°C)** | **Between groups** | 56.14 | 2 | 28.07 | 10.44 | <0.01 |
| | **Within group** | 126.40 | 47 | 2.69 | | |
| | **Total** | 182.54 | 49 | | | |
| **Conductivity (μS/cm)** | **Between groups** | $2.13 \times 10^9$ | 2 | $1.07 \times 10^9$ | 11.04 | <0.01 |
| | **Within group** | $4.54 \times 10^9$ | 47 | $9.66 \times 10^7$ | | |
| | **Total** | $6.67 \times 10^8$ | 49 | | | |

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
