# Peer review of "The Spatial Distribution and Morphological Characteristics of Chum Salmon (Oncorhynchus keta) in South Korea"

_fishes, doi:10.3390/fishes7010027_

Round 1

Reviewer 1 Report

General comments

The manuscript addresses the study of a salmon population and its distribution in water bodies in various areas in South Korea, taking into account the physical and chemical aspects of the water. It also assesses the distribution of length-weight relationships and sex ratio of the species. The authors had access to a lot of data provided by various institutions over several years.

Regarding the form the manuscript (ms) lacked greater care as to the content of the paragraphs, which mix different subjects and deserve a better division into different paragraphs. There are errors in the mentions of figure numbers in the text (see specific comments). There are English errors so the ms needs to be proofread by an English-speaking person. In the discussion, there are parts of results and an initial paragraph that would be the introduction. It is necessary to standardize throughout the text and in the figures the abbreviations of DO, BOD, TN, TP, which are usually written this way, while the ms has several different abbreviations. There is a lot of repetition of chum salmon in the text that in many cases could be replaced by the words “individuals and species”.

The maps are not detailed enough to show the water bodies and the connection between them. Two dams are mentioned, but the discussion does not go into depth on their role on the characteristics of the salmon population and how they affect the migration. Water characteristics are important as evidenced in the PCA and correlation table, but there is no mention in the ms of values ​​and how they differ between the areas (they could be included as supplementary data), so that the areas are not characterized regarding pollution. BOD is an important measure of water quality and is often used to compare organic matter pollution, but there is no evaluation of it. What is needed, then, is a better characterization of the stretches under study in terms of pollution levels, since DO concentration is a very important factor for the species. The migration route is not explored and the reader is not clearly informed as to how this happens in the various stretches studied. For example, does area R1 correspond to the end of the species migration? Which change has occurred so far to justify the conclusion about the influence of global warming? Are there data on whether the pH of the marine region has decreased, for example?

The authors have a lot of data, which if well explored and combined with greater care with the text can greatly improve the ms.

Some specific comments

Line 12: check if Taiwan is correctly written.

Line 36: replace “salmonids are comprised of 11 genera” by “salmonids comprise 11 genera”.

Line 195: figure 2 instead of figure 3.

Line 220: figure 4 instead of figure 2.

Lines 243-245: sentence out of context as it is part of the introduction.

Line 271: example of results in the discussion.

Line 272: figure 4 instead of figure 5.

Author Response

Manuscript ID: fishes-1475541

First of all, the co-authors and I would like to appreciate to reviewers and editor for their kind response and comments to improve our manuscript draft (fishes-1475541). According to the comments, we would like to submit a written reply. Responses (corrections) for Reviewer 1 are colored in red.

  1. Regarding the form the manuscript (MS) lacked greater care as to the content of the paragraphs, which mix different subjects and deserve a better division into different paragraphs.
  • We agree that there are some statements that should be rearranged. Specifically for line 247-249 (in original manuscript), which is about the chum salmon’s migration route to Korea, should be rearranged to introduction part, because it is giving a basic information of the species. Therefore, we rearranged line 247-249 to the middle of introduction (line 58-60, in the revised manuscript) for better context.

  1. There are errors in the mentions of figure numbers in the text (see specific comments).
  • We revised the errors in the mentions of figure numbers in the text, as mentioned in specific comments (please see the specific comments 3, 4 and 7).

  1. There are English errors so the MS needs to be proofread by an English-speaking person.
  • The article have been revised by English proofreading company (Editage Co., https://www.editage.co.kr/, Document ID: PSUS_6476). Also, Maurice JM Lineman, one of the co-authors (native Canadian) revised this article several times when submitting this revision.

  1. In the discussion, there are parts of results and an initial paragraph that would be the introduction.
  • As we already mentioned in the first reply, we rearranged lines 247-249 to the middle of the introduction (line 58-60) for better context.

  1. It is necessary to standardize throughout the text and in the figures the abbreviations of DO, BOD, TN, TP, which are usually written this way, while the MS has several different abbreviations.
  • We revised abbreviations D.O, B.O.D, T.N and T.P => DO, BOD, TN and TP.

  1. There is a lot of repetition of chum salmon in the text that in many cases could be replaced by the words “individuals and species”.
  • We appropriately replace chum salmon to “Individuals, species, they or it” or deleted the words if unnecessary throughout the manuscript.

  1. The maps are not detailed enough to show the water bodies and the connection between them. Two dams are mentioned, but the discussion does not go into depth on their role on the characteristics of the salmon population and how they affect migration.
  • We revised the maps to show the water bodies clearly and the connection between them, and stated a sentence to explain there are two estuarine barrages in the mouth of the Nakdong River (line 103-104, Fig. 1). Also, we rephrased the sentence to emphasize the important role of dams (line 301-305).

  1. Water characteristics are important as evidenced in the PCA and correlation table, but there is no mention in the MS of values ​​and how they differ between the areas (they could be included as supplementary data) so that the areas are not characterized regarding pollution.
  • We supplemented two tables on water qualities and how they are different according to different regions by using One-way ANOVA. The two tables are presented in the appendix (page 14-15, Table A-5, 6). Also, we added some sentences that describe the results of water quality in the result session (line 219-224).

  1. BOD is an important measure of water quality and is often used to compare organic matter pollution, but there is no evaluation of it. What is needed, then, is a better characterization of the stretches under study in terms of pollution levels, since DO concentration is a very important factor for the species.
  • Including the statement in the question 8, we added an additional discussions, in terms of pollution levels (line 281-282).
  1. The migration route is not explored and the reader is not clearly informed as to how this happens in the various stretches studied. For example, does area R1 correspond to the end of the species migration?
  • In this research paper, we mainly focus on the inland spatial distribution and migration, therefore little information were prepared in exploring the migration routes, or finding their final destinations. Further studies on migration routes would be proceeded in the next research.

  1. Which change has occurred so far to justify the conclusion about the influence of global warming?
  • In the PCA performed, we are able to observe a negative correlation of Catch with Temperature. In this point of view, it could be understood that river environment with high temperature could display low catch. Therefore, global warming could cause decrease in abundance. We therefore added a sentence (line 280-282) to emphasize the potential effects of global warming on the chum salmon abundance.

  1. Are there data on whether the pH of the marine region has decreased, for example?
  • Although marine water quality could play an important role in salmon life history, we only utilized inland water quality indices, which leaves a limitation to fully discover the correlation with the catch. Therefore, we added the limitations in this study (line 325-328).

Specific comments 

  1. Line 12: check if Taiwan is correctly written.
  • Taiyuan in this text is not referring to Taiwan. Taiyuan is city of Shanxi Province, in People’s Republic of China. Therefore, I would like to leave it unchanged.

  1. Line 36: replace “salmonids are comprised of 11 genera” by “salmonids comprise 11 genera”.
  • We have revised according to reviewer’s comment (line 36)

  1. Line 195: figure 2 instead of figure 3
  • We have revised according to reviewer’s comment (line 202)

  1. Line 220: figure 4 instead of figure 2
  • We have revised according to reviewer’s comment (line 230)

  1. Lines 243-245: sentence out of context as it is part of the introduction.
  • We have deleted the line 243-245, as it was already mentioned early.

  1. Line 271: example of results in the discussion
  • I believed that it would be better if I refer the result once more to emphasize the importance of our results. However, since repetition of our results could impede the understanding of the context, we decided to delete the line 271 for better formations.

  1. Line 272: figure 4 instead of figure 5
  • We have revised according to reviewer’s comment (line 273)

Reviewer 2 Report

The subject is very interesting and has a positive contribution to the literature, especially for future surveys in fish spatial distribution. The authors made a huge effort to compile a large data set of biological and environmental data. The findings were handled and reported in a way that fully revealed the purpose of the study. I consider the basic premise, method developed and questions asked by this study as valuable and interesting. Tables and figures are also well presented and titled. The discussion is detailed enough and the scope of the ms very well presented with high quality of reference and examples.
Concluding, the ms could be proceeded for publication after minor edits and comments taken into consideration.

In lines 145 and 146 the authors mentioned some details on the period of the study (2004-2020), however, there is not a clear declaration of the period of the study. This would be also feasible for the other parameters too, apart for the fish morphometrics.

The period of the study and the number of samples would be nice to be mentioned in the abstract.

Author Response

Manuscript ID: fishes-1475541

First of all, the co-authors and I would like to appreciate to reviewers and editor for their kind response and comments to improve our manuscript draft (fishes-1475541). According to the comments, we would like to submit a written reply. Responses (corrections) for Reviewer 2 are colored in blue.

  1. In lines 145 and 146 the authors mentioned some details on the period of the study (2004-2020), however, there is not a clear declaration of the period of the study. This would be also feasible for the other parameters too, apart for the fish morphometrics.
  • We supplemented a statement of utilized years of water quality (line 127-129)

  1. The period of the study and the number of samples would be nice to be mentioned in the abstract.
  • We supplemented a statement of period of study and the number of samples (line 20-23)

Round 2

Reviewer 1 Report

Comments on the revised version of the manuscript fishes-1475541
The manuscript has improved regarding the first version. However, there are still some things to be fixed as specified below. Much has been said about the effect of global warming on organisms and ecosystems, but in order for this statement not to be just speculative in the manuscript, it is necessary to include some data on the temperature limits tolerated by this species of salmon and how the increase of 1.1oC on average that occurred until now (see IPCC report), as well as forecasts for the coming decades, can affect it. Another point that draws attention and deserves to be commented on is the high conductivity in R4. The reader would like to be informed of the reason for such a high value, as other abiotic factors do not answer this question, as there is no evidence of high pollution (low BOD, for example).
Specific comments
Table 1: Table 1: species name in italics.
Line 191: replace “species” by “specimens”.
Lines 228-229: replace “pH and DO is” by “pH and DO are”.
Lines 242-245: rewrite the sentence.
Line 327: replace “data was collected” by “data were collected”.
Line 337: replace “they” by “it”, as it refers to one species.
I suggest “minor revision”.

Author Response

Manuscript ID: fishes-1475541

First of all, the co-authors and I would like to appreciate to reviewers and editor for their kind response and comments to improve our manuscript draft (fishes-1475541). According to the comments, we would like to submit a written reply. Responses (corrections) for Reviewer 1 are colored in red.

Reviewer 1

  1. Much has been said about the effect of global warming on organisms and ecosystems, but in order for this statement not to be just speculative in the manuscript, it is necessary to include some data on the temperature limits tolerated by this species of salmon and how the increase of 1.1oC on average that occurred until now (see IPCC report), as well as forecasts for the coming decades, can affect it.
  • The optimal temperature for chum salmon is known to be around 7 to 11, while its upper lethal temperature cannot be extended beyond 24. Also, a decrease in carrying capacity, loss of migration routes for chum salmon have been occurred and predicted. We supplemented this information (line 287-292) to emphasize the importance of temperature.

  1. Another point that draws attention and deserves to be commented on is the high conductivity in R4. The reader would like to be informed of the reason for such a high value, as other abiotic factors do not answer this question, as there is no evidence of high pollution (low BOD, for example).
  • The high conductivity in R4 is due to the water quality measuring station being located in a brackish zone, exhibiting salinity around 15psu. We supplemented this information (line 282-284) for better understanding.

Specific comments from Reviewer 1

  1. Table 1: Table 1: species name in italics in title
  • We changed species name into italics (Table 1., line 176).

  1. Line 191: replace “species” by “specimens”.
  • We changed species to specimens (line 185)

  1. Lines 228-229: replace “pH and DO is” by “pH and DO are”.
  • We changed ‘is’ to ‘are’ (line 222-223)

  1. Lines 242-245: rewrite the sentence.
  • We rewrote the sentence for better understanding (line 231-238 and line 241-243).

  1. Line 327: replace “data was collected” by “data were collected”.
  • We replaced ‘data was collected’ by ‘data were collected’ (line 315).

  1. Line 337: replace “they” by “it”, as it refers to one species.
  • We replaced ‘they’ by ‘it’ (line 325).